# The Level of COVID-19 Anxiety among Oncology Patients in Poland

**DOI:** 10.3390/ijerph191811418

**Published:** 2022-09-10

**Authors:** Mateusz Grajek, Karolina Krupa-Kotara, Mateusz Rozmiarek, Karolina Sobczyk, Eliza Działach, Michał Górski, Joanna Kobza

**Affiliations:** 1Department of Public Health, Faculty of Health Sciences in Bytom, Medical University of Silesia in Katowice, 40055 Katowice, Poland; 2Department of Epidemiology, Faculty of Health Sciences in Bytom, Medical University of Silesia in Katowice, 40055 Katowice, Poland; 3Department of Sports Tourism, Faculty of Physical Culture Sciences, Poznan University of Physical Education, 61871 Poznan, Poland; 4Department of Health Economics and Health Management, Faculty of Health Sciences in Bytom, Medical University of Silesia in Katowice, 40055 Katowice, Poland; 5Department of Dietetics, Faculty of Health Sciences in Bytom, Medical University of Silesia in Katowice, 40055 Katowice, Poland

**Keywords:** COVID-19, SARS-CoV-2, oncology, anxiety, FCV-19S, GAD-7, Poland

## Abstract

Cancer patients tend to have a high psychological burden. Half of cancer patients suffer from severe affective disorders and anxiety disorders, while one-third struggle with mild forms of these. The COVID-19 pandemic is damaging the mental health of the population due to social restrictions. A growing number of studies note the role of COVID-19 anxiety in the health and quality of life of cancer patients. The purpose of this study is to estimate the level of COVID-19 anxiety among oncology patients and to test the utility of the FCV-19S scale in a population study of cancer patients. The study included 600 respondents (300 oncology patients and 300 control subjects not undergoing oncological treatment). The FCV-19S scale and the GAD-7 scale were used in the study. The results were interpreted according to the following verbal scale: 76–100%, high anxiety; 56–75%, moderate anxiety; 26–55%, low COVID-19 anxiety; <25%, no COVID-19 anxiety. In the analysis of the GAD-7 questionnaire results, the mean score obtained was 8.21 (min. 0; max. 21; SD 5.32). For 81% of respondents in the group of oncology patients, the total score indicated the presence of anxiety symptoms with varying degrees of severity; in the control group, this proportion was 55% of respondents. The FCV-19S scale score as a percentage was 57.4% for oncology patients, indicating a moderate level of fear of the SARS-CoV-2 virus, and 30.3% for the control group, indicating a low level of fear of the SARS-CoV-2 virus. One-fifth of oncology patients were afraid of losing their lives due to the SARS-CoV-2 virus; in the control group, this proportion was 13% of respondents. Oncology patients were characterized by a higher prevalence of sleep disturbance than control group respondents, which was associated with greater anxiety. The study, therefore, shows that oncology patients have moderate levels of anxiety associated with the COVID-19 pandemic, and non-oncology patients show lower levels of anxiety.

## 1. Introduction

Anxiety can be experienced as a feeling of nervousness, tension, emotional distance, worry, or fear that something bad is about to happen. It can also cause physical symptoms such as shortness of breath, rapid heartbeat, sweating, cold hands, trembling, trouble concentrating, and difficulty sleeping [1,2]. It is important to remember that anxiety is experienced and described uniquely in different cultures and may be named differently [3]. Reactions to stressful events are normal human sensations. Extremely stressful situations, including the current COVID-19 outbreak, can lead to strong feelings of anxiety, which can induce stress and prevent normal functioning [4]. Data from around the world indicate that anxiety about the future associated with the COVID-19 pandemic was heightened among people using healthcare facilities. Thus, in the current study, we decided to focus on a group of chronically ill people—oncology patients [5,6].

Cancer is a global health problem and one of the leading causes of death worldwide [7,8]. Cases of coexistence of cancer and SARS-CoV-2 infection have only been described in preliminary scientific reports, and evidence from large population-based studies testing whether cancer and treatment-related therapy exacerbate the risk of COVID-19 infection is still limited [8]. According to a study by Liang et al. [9], the rate of COVID-19 infection among cancer patients is higher than in the general Chinese population. On this basis, it can be suspected that cancer patients are more vulnerable to SARS-CoV-2 infection. In addition, a retrospective study by Zhang et al. [10] including cancer patients with COVID-19 from three hospitals in Wuhan, China, demonstrated the unique susceptibility of this group to infection with the new coronavirus. As demonstrated by Lee et al. in a large population-based trial that included more than 20,000 cancer patients, there is a significantly higher risk of COVID-19 infection among cancer patients, especially among the elderly and men [2]. Individuals with cancer are generally more susceptible to infection due to treatment with immunosuppressive agents and chemotherapeutics [11].

Cancer patients also tend to have a high psychological burden [12,13,14,15]. A significant portion (50%) of cancer patients suffer from severe anxiety and mood disorders, while 30% struggle with their mild forms [16]. The COVID-19 pandemic affected the mental health of the population due to the social restrictions that were systematically introduced [16,17,18,19,20]. The COVID-19 pandemic resulted in the closure of many support centers, foundations, and associations where patients could exchange views and insights on treatment [21]. It should also be noted that patients were often left alone with their disease, cut off from their families, as there was a total ban on visits to hospital wards, which caused a drop in mood and affected the development of depressive disorders [15]. All these factors could negatively affect the success of further treatment [22].

An increasing number of authors are choosing to study the impact of the COVID-19 pandemic on people’s mental states [1,2,3,4,5,6]. Studies have shown an increased psychological burden related to stress, anxiety, and depression since the onset of COVID-19 restrictions [23]. Brooks et al. [24] studied the psychological effects of quarantine during the pandemic, indicating the psychological burden on individuals who are unable to participate in public life. Similarly, Grajek and Bialek-Dratwa’s study showed that the pandemic situation has a significant impact on the deterioration of the mental state of cancer patients [25]. Several studies have highlighted the correlation between increased fear and anxiety towards COVID-19 and psychophysical deterioration [26,27,28,29,30,31,32]. It is also worth mentioning that some WHO reports indicate the existence of symptoms similar to post-traumatic stress disorder in the population [33,34,35].

Given the severe global threat and the impact that the COVID-19 pandemic has had on various aspects of life, Ahorsu et al. [28] developed a scale to measure COVID-19 anxiety, the Fear of Coronavirus-19 Scale (FCV-19S). This scale has been used in many countries such as Iran [36], Bangladesh [37], Italy [38], Turkey [39], Russia and Belarus [40], Israel [41], Peru [42], and Paraguay [43]. In the Polish literature, the usefulness of the scale was confirmed by the authors Pisula and Nowakowska [44]. However, the validity of this scale has not been compared with other known anxiety scales, e.g., GAD-7, the General Anxiety Disorder Scale.

We aimed to estimate the level of general and COVID-19 anxiety among oncology patients. Our research hypothesis was that cancer patients represent a group at increased risk of having anxiety disorders, including COVID-19 anxiety. The essence of the study implies the use of the GAD-7 and FCV-19S scales to estimate whether the anxiety present among patients is due to the epidemiological situation or whether this is a group of people at greater overall risk of having generalized anxiety disorder.

## 2. Materials and Methods

### 2.1. Participants

The study group (SG) consisted of 300 patients treated in an oncology outpatient clinic and 300 patients in the control group (CG), who were patients of a general outpatient clinic. The gender (t = 6.543; *p* = 0.338) and age (t = 4.311; *p* = 0.298) structure did not differ significantly between the groups, which shows the cohesion of the groups: 62% of the oncology patients (n = 168) and 57% of the control group (n = 171) were women. The mean age of the oncology patients was 42 (±5) years and that of the control group was 39 (±6) years. More than three-quarters of the participants lived in a city (78%) and 22% lived in a village.

### 2.2. Sample Size Estimation

It was estimated that a sample of 300 oncology patients would be sufficient and representative of the Silesian region in Poland. It was assumed, according to the National Cancer Registry [45] and Health at a Glance 2021 OECD report [46], that the cancer incidence rate in Poland is 267 patients per 100,000, resulting in more than 101,000 cancer patients in the country and 12,000 in Silesia. Accordingly, we used the following formula: N_min_ = NP · (α^2^ · f(1 − f)) ÷ NP · e^2^ + α^2^ · f(1 − f), where N_min_ is the minimum sample size, NP is the size of the population from which the sample is taken, α is the confidence level for the results, f is the size of the fraction, and e is the assumed maximum error. The minimum sample size of oncology patients for the Poland and Silesian population was calculated as 138 people (α = 0.95; f = 0.9; e = 0.05). Based on these calculations, the collected group of oncology patients was considered representative, and the control group (non-oncology patients) was selected so that both groups were equal for ease of calculation and presentation of results.

### 2.3. Eligibility Criteria

The criteria for inclusion of individuals in the study assumed that respondents had no history of ongoing psychological care and/or psychiatric treatment (not including psychological counseling related to the oncology pathway). All respondents entered the study voluntarily and anonymously. Respondents were selected based on independent randomization performed by the oncology coordinator, and for those outside the oncology patient group, randomization was based on current patient registration. For the study, only data on gender, age, and the fact of oncology treatment or lack thereof were collected. Respondents were familiarized with the method of measurement. Each participant received an Internet form with the FCV-19S scale and the GAD-7 to fill out (it was decided to survey via the Internet due to sanitary restrictions related to the current epidemiological situation). Consenting subjects participated in the study. The study complies with the provisions of the Helsinki Declaration. The study in light of the Act of 5 December 1996, the professions of doctor and dentist (Journal of Laws of 2011, No. 277, item 1634, as amended), is not a medical experiment and did not require the approval of the Bioethics Committee of the Medical University of Silesia in Katowice.

The composition of the sample (SG and CG groups) is shown in Figure 1.

### 2.4. Measurements

We used the GAD-7 screening questionnaire to determine feelings associated with generalized anxiety disorder (GAD). It is a 7-item scale, based on a 4-point Likert scale, used to assess the level of anxiety, as well as to assess the risk of generalized anxiety syndrome. The questions allow the respondents to assess their feelings of anxiety, tension, and nervousness, along with their ability to control these feelings, the ease with which they arise, and problems with relaxation. For each question, respondents earn 0 to 3 points depending on the frequency of occurrence of a given phenomenon (0—not at all; 1—a few days; 2—more than half of the days; 3—almost every day) within the past 14 days. Scores of 5, 10, and 15 points indicate mild, moderate, and severe anxiety, respectively. A score of at least 10 points indicates a high probability of generalized anxiety syndrome [47]. Cronbach’s α was 0.88, which indicates very good reliability.

Moreover, the FCV-19S scale was used according to Ahorsu et al. [28] (Polish translation—Pisula and Nowakowska, 2020 [44]). Subscales are scored according to Likert scale assumptions, where 1 means full disagreement with the statement and 5 means full agreement with the statement. The scale consists of 7 items, and the maximum possible score is 35 points. To obtain the percentage value necessary for data analysis, the raw score was multiplied by the value of the calculated coefficient (≈2.857; the value of the coefficient was calculated based on the mathematical operation—100%/maximum possible score (35)). A verbal interpretation of the results obtained was adopted for the study: 76–100% indicates high COVID-19 anxiety; 56–75% indicates moderate anxiety; 26–55% indicates low anxiety; <25% indicates no COVID-19 anxiety. Cronbach’s α was 0.92, which indicates very good test reliability.

### 2.5. Statistical Analysis

Kruskal–Wallis and Mann–Whitney U tests were used in the statistical processing of the data. The probability level was 0.05. The study was preceded by a pilot study, conducted on a group of 30 patients, where respondents could assess whether they understood the questions contained in the questionnaire.

## 3. Results

In the analysis of GAD-7 questionnaire results, the mean score was 8.21 (MIN = 0; MAX = 21, SD = 5.32). For 81% of respondents in the group of oncology patients, the total score indicated the presence of anxiety symptoms of varying degrees of severity, while in the control group, this proportion was 55% of respondents. Moreover, 44% and 11% of respondents from the group of oncology patients and the control group received a score of at least 10 points, respectively, which is the basis for the probability of having generalized anxiety syndrome. The exact results are shown in Table 1.

In the last 14 days before the survey, 86% of the entire sample had experienced irritability, anxiety, and tension, with one in five respondents experiencing these conditions almost daily. Additionally, a significant percentage (21%) of respondents had difficulty relaxing daily. During the analysis, a close correlation was determined between the perception of anxiety and the study group and gender. Those undergoing oncological treatment received higher scores in the questionnaire compared to the control group (81% vs. 55%); this relationship was confirmed by statistical tests (*t = 13.341*; *r = 0.725*; *p = 0.001*). In addition, women scored significantly higher on the GAD-7 compared to men (*t = 11.846*; *r = 0.653*; *p = 0.023*). There was no correlation between anxiety prevalence and age.

The portion of the study that used the FCV-19S scale showed that 24% of oncology patients and only 3% of the control group reported that they strongly agreed with the statement: “I am very afraid of the SARS-CoV-2 virus”. Among cancer patients, the most frequently selected option was “rather agree” (35%), and the arithmetic mean of the responses was 4.1 points. People who have not received oncological treatment most often chose the answer “I have no opinion” (43%), and the arithmetic mean in this group was 2.8 points (Table 2 and Table 3).

The second question of the questionnaire concerned feelings of anxiety at the thought of the SARS-CoV-2 virus. Thirty-two percent of oncology patients strongly agree that they feel such anxiety, and the highest number of points was given in this group with the highest frequency, while in the control group, eleven percent of respondents marked such an answer (Table 2). The arithmetic mean for this question was 3.5 points among oncology patients and 2.2 points in the control group (Table 3).

The statement “My hands sweat when I think of coronavirus” was strongly disagreed with by 23% of oncology patients and more than twice as many in the control group, 48% (Table 2). The arithmetic mean of oncology patients, in this case, was 1.9 points, and in the control group, 1.2 points (Table 3).

One-fifth of the respondents in the oncology treatment group were afraid of losing their lives due to the SARS-CoV-2 virus, compared to 13% of the respondents in the control group (Table 2). The arithmetic mean of responses for the statement “I fear a loss of life due to coronavirus” was 2.5 points for oncology patients and 0.9 points in the control group (Table 3).

A greater impact of pandemic information from social media was also observed among those receiving cancer treatment compared to the control group. Almost a quarter of the respondents in the first group (24%) strongly agreed with the statement, “When I watch the news and learn about coronavirus-related stories on social media, I get nervous or feel anxious”, compared with only 3% of respondents in the control group (Table 2). The arithmetic means for this question were 3.3 points for oncology patients and 1.1 points for the control group (Table 3).

Nineteen percent of people undergoing oncology treatment strongly agreed with the statement that they could not sleep for fear of SARS-CoV-2 infection, whereas in the control group, three people (1%) marked this response. However, for this statement, the most frequently selected option in both groups was “strongly disagree” (31% and 42%) (Table 2). The mean response to this statement was 2.5 points in the oncology patient group and 1.1 points in the control group (Table 3).

The last statement of the questionnaire was “My heart beats rapidly when I think of coronavirus infection”. Twenty-two percent of the oncology treatment subjects strongly agreed with this statement, along with only one percent of the control group. Again, respondents in both groups were most likely to select the “strongly disagree” response (Table 2). The arithmetic means for this statement were 2.3 points for oncology patients and 1.1 points for non-oncology patients (Table 3).

The FCV-19S total scale score as a percentage was 57.4% for oncology patients, indicating moderate SARS-CoV-2 anxiety, and 30.3% for the control group, indicating low SARS-CoV-2 anxiety. Intergroup comparisons showed that the GAD-7 scores for generalized anxiety overlapped with the FCV-19S scores for moderate anxiety—subjects scoring above 10 on the GAD-7 on the FCV-19S scale scored above 20 points (SG) and 10 points (CG), respectively (*t = 11.299*; *r = 0.678*; *p = 0.001*). It has been observed that compared to FCV-19S, GAD-7 yields higher results, especially among oncology patients.

For a better idea of the study participants’ anxiety, they were asked to name the most worrisome factor related to the ongoing pandemic that makes them feel anxious. In both the SG and CG groups, most people feared the possibility of becoming sick and dying alone (82% SG and 54% CG). Subsequently, respondents reported issues such as the possibility of complications (46% of SG and 56% of CG), delay of planned treatment (62% of SG and 38% of CG), the possibility of losing a source of income (28% of SG and 32% of CG), and separation from loved ones (36% of SG and 26% of CG) (Table 4).

The obtained reasons for anxiety among the patients studied were correlated with the highest scores on the GAD-7 and FCV-19S scales to determine whether the same reasons for anxiety could lead patients to achieve high scores on both scales. Based on the assessment, it was found that those characterized by generalized anxiety (GAD-7) in the SG group most often indicated isolation from family (52%) as a reason for anxiety, while for those in the CG group, a reason for anxiety was a loss of income (54%). In the case of the FCV-19 scale, those with the highest anxiety scores indicated a delay in planned therapy as the reason for anxiety, and this reason is the most frequently selected in both study groups (*t = 9.983*; *r = 0.641*; *p = 0.020*).

At the end of the conducted research, respondents were asked whether they receive psychological support at their own or other healthcare facilities and whether they have ever used it. In this regard, 52% of oncology patients and 44% of those in the control group said they would be happy to receive such support during treatment, but they had never been informed of such an opportunity. In addition, 88% and 86% of respondents (SG and CG, respectively) indicate that there is no working position for a psychologist or other mental health professional at their facility. It is worth noting at this point that a positive relationship was observed between the level of anxiety measured by both the GAD-7 and FCV-19S scales and the need for psychological help. Those with higher levels of anxiety were more likely to indicate the need for such a specialist (psycho-oncologist) at their facility (*t = 10.112*; *r = 0.512*; *p = 0.030*). No differences were found between the groups of oncology and non-oncology patients.

## 4. Discussion

Cancer patients are at risk for many mental health disorders, and anxiety is one of the most frequently cited burdens they face. Research on the relationship between anxiety and depression in cancer patients indicates a high prevalence of these disorders in adults with a cancer diagnosis [47,48]. Anxiety is a part of cancer and accompanies many patients from the time of diagnosis through the process of treatment, remission, or recurrence of the disease to the time of death [49,50,51,52]. The most common fear is the fear of progression (FoP), which is the fear of disease progression or recurrence. It is a phenomenon that can affect all chronic diseases, including cancer [53]. In this study, we also noted that one reason for anxiety may be delays related to the planned treatment. Interestingly, this reason was given not only by cancer patients, which would indicate that this predictor is also important in other populations. However, it should be remembered that a specific period such as the COVID-19 pandemic also significantly alters the reasons for anxiety. This is indicated by other studies and reports cited below.

The omnipresent fear and uncertainty associated with the epidemiological situation have not spared Polish oncology. According to the report “Oncology 2025” [54], there has been a huge decrease in performed mammography and cytology examinations during the pandemic. The number of patients diagnosed with cancer or suspicion of cancer also declined in comparison to the previous year. Analyzing the absolute numbers, the decrease is up to 30%. Consequently, there were 25% fewer multidisciplinary specialist meetings. Analyzing first-time services in oncologic surgery, the first decreases were already noted in March 2020, while May 2021 saw the largest decrease compared to last year. In the field of clinical oncology, about a quarter fewer patients appeared. The provision of drug programs in oncology also fell by several percentage points. First-time consultations for radiotherapy fell by about 30% compared to the previous year, and a similar decline was noted in the delivery of radical and palliative radiotherapy services [55].

The reasons for the situation in Poland can be attributed to the efforts of the government to limit the transmission of the SARS-CoV-2 virus in the population, as well as the transformation of many hospitals into so-called “one-name hospitals” (commonly known as COVIDs). Suddenly introduced measures such as isolation, restriction of civil liberties (limitation of social contacts, prohibition of movement), and risk of virus infection may lead to anxiety disorders or depression. A study by Babicki et al. [56] involving 2457 Poles confirmed that the COVID-19 pandemic causes anxiety, increased feelings of apprehension, and concern about the future. Moreover, anxiety disorder tendencies in the study group were more frequent in women than in men. In oncology patients, levels of anxiety and psychological distress may also be influenced by awareness of the increased risk of SARS-CoV-2 infection in this patient group [57]. An additional factor is the dissemination of information by the mass media regarding the higher mortality rate in patients with chronic diseases [58].

Scientific reports have found information about patients postponing medical appointments due to fear of SARS-CoV-2 virus infection. In a study by Fujita et al. [59] involving 165 patients treated at the National Hospital Organization Kyoto Medical Center, 9.1% of lung cancer patients experienced anxiety and chose to delay treatment during the COVID-19 pandemic, while the work of Karacin et al. [57], which analyzed the records of 3661 patients undergoing chemotherapy, proved that the rate of treatment delays increased significantly (from 11.6% to 14.2%) during the COVID-19 pandemic. Among the reasons for postponing chemotherapy was fear of the pandemic, with 13.6%, and after the introduction of telemedicine, this percentage decreased to 4.6%. The study by Vanni et al. [60] included 78 breast cancer patients from Tor Vergata University Hospital in Rome. The group was divided into women enrolled for surgery before and after the first case of SARS-CoV-2 in Italy. In the first group, 9.3% of the women were refused, while in the second group, as many as 35.9% of the study group were refused.

A cross-sectional study by Ng et al. [61] conducted in Singapore included 624 oncology patients, 408 caregivers of these patients, and 421 healthcare professionals. The latter group was less fearful of COVID-19 compared with patients and caregivers of patients, with 66.0% of patients and 72.8% of caregivers experiencing severe or extreme fear of SARS-CoV-2 compared with 41.6% of healthcare workers. Similar results were obtained by Gianluca et al. [60], and also confirmed by data published in the Oncology Portal [62].

Our study builds on the topic of fear of SARS-CoV-2. The results confirm that the level of this fear is significantly higher in oncology patients compared to the control group of people not undergoing oncology treatment. Sigorski et al. [63] conducted a study with a similar theme. The authors determined the level of fear of the SARS-CoV-2 virus among patients in five Polish oncology centers. A total of 306 patients participated in this study. The mean FCV-19S score in this group was 18.5 ± 7.44. A similar mean FCV-19S score of 18.48 ± 5.32 was obtained by Parlapani et al. in their study [64]; however, their work involved elderly people living in Greece.

Awareness of specific COVID-19 exposure among chronic patients may influence the emergence of fear of contracting the disease. Korukcu et al. [65], in their study conducted on the Turkish population, proved that the FCV-19S scale score was significantly higher in people with chronic diseases and using long-term medication (20.4 ± 6.8) compared to healthy people (19.7 ± 7.3). In the study by Tokgoz et al. [66], which included women treated at a fertility center in Turkey, the mean FCV-19S score was 16.7 ± 5.3. In addition, the authors noted that women who wanted to postpone treatment for the pandemic had higher levels of COVID-19 anxiety (17.7 ± 4.7) than women who wanted to continue treatment (14.3 ± 5.7). This suggests that some patients may postpone treatment for chronic diseases due to fear of contracting COVID-19. The publication by Rahman et al. [67] includes a study of SARS-CoV-2 anxiety among 587 Australian adults. In this work, the mean FCV-19S scale score was 18.4 (±6.5). The mean FCV-19S score in a study by Giordani et al. [68] involving 7430 Brazilians was 19.8 ± 5.3. The authors also demonstrated that anxiety levels were higher in individuals who used more protective measures (e.g., wearing a mask) and maintained social distancing. In the Indian study by Doshi et al. [69], the means for all statements of the FCV-19S scale were high.

The development of the COVID-19 pandemic should undoubtedly be considered one of the greatest stressors affecting the general population. The disease has affected the quality of life and the way different communities around the world function daily, regardless of economic or cultural factors. The value placed on life can exacerbate the occurrence of anxiety caused by the vision of loss of health or death. This anxiety may be particularly exacerbated in the elderly, chronically ill, and cancer patients, as demonstrated in this study [70]. Given the results obtained, it is undeniably important for oncology patients facing health problems and stressful epidemiological situations to equip medical facilities with psychological support in the form of a specialized clinical psychologist, psychotherapist, or psycho-oncologist [71]. Professional emotional and psychological support play an important role in the comprehensive treatment of an oncology patient [72]. An experienced psycho-oncologist can help a patient cope with a difficult diagnosis and facilitate the patient’s path through treatment [73]. The support of a psycho-oncologist is important at every stage of cancer [72]. Experts often stress the importance of the so-called interdisciplinary and holistic approach to cancer treatment in modern oncology [73]. The role of a mental health specialist in the lives of oncology patients is significant in light of the findings of the study, where nearly 90% of patients could not count on such support.

On the practical and managerial side, the research conducted indicates that the phenomenon of anxiety is present in the population of cancer patients, which is normal, but the severity of this phenomenon is significantly increased by an additional external cause (such as the COVID-19 pandemic). This can be inferred by comparing the two study groups. The conclusions of the study are particularly useful for medical nursing practice and present the need for further measures to prevent the development of psychiatric and psychosomatic disorders underpinned by anxiety, fear, stress, and trauma. Oncology patients, as a specific group of emergency patients, should receive psychosocial support in the form of psychological and psychotherapeutic consultations from the very beginning of their diagnostic and treatment path. Although the current study did not indicate high levels of anxiety measured with either tool, it is important to keep in mind any confounding factors mentioned in the study’s limitations. In addition, it must be emphasized that the epidemiological situation related to COVID-19 is still unclear, and it is unclear what mental health implications it will bring. Further research should focus on evaluating the FoP phenomenon and discussing detailed data based on clinical cases.

## 5. Strengths and Limitations

The strength of the conducted study is the large group on which the questionnaire was conducted. In addition, the use of standardized tools and comparison of obtained results allowed for a better understanding of the nature of anxiety associated with the risk of SARS-CoV-2 infection. The results of the study provide valuable information from both the theoretical and practical sides.

Unfortunately, the study also failed to avoid errors, which should be emphasized. First of all, the questionnaire was conducted via the Internet (CAWI method), which, despite its widespread use, still arouses much controversy. Such an approach was dictated by the period of the study, i.e., the period of the highest number of COVID-19 infections in Poland. Such an approach can also be justified by the fact that the authors made every effort to minimize the risk of bias—the study began with a pilot study, methods were used to avoid the phenomenon of “bot/fake responders”, and the study was coordinated by researchers experienced in contact with patients.

It should also be noted at this point that the results of the study may have been influenced by various factors that were not considered during the planning of the study, but which may prove important to investigate in subsequent studies. Such factors include the type of cancer, length of treatment, duration of psychologist coverage, and selected socioeconomic issues. All of these are worth expanding on in future research projects.

## 6. Conclusions

Oncology patients have a moderate level of anxiety related to the presence of the COVID-19 pandemic. Non-oncology patients show a lower level of anxiety about COVID-19 (for the FCV-19S scale). According to the results of the GAD-7 questionnaire, in the study population, the group of people at risk of generalized anxiety syndrome mainly comprises oncology patients. Determination of the level of pandemic anxiety can contribute to the application of appropriate methods of assistance to people with moderate and high levels of anxiety. In addition, it is worth noting that there is a certain percentage of oncology patients who do not feel anxiety related to the epidemiological situation. This fact should be studied more extensively with the consideration of more variables and could be attributed to the external support that such individuals are likely to receive, which influences the lower level of COVID-19-related anxiety they feel. This emphasizes the role of cooperation of the patient’s loved ones with the help of qualified psychological and psychiatric staff.

## Figures and Tables

**Figure 1 ijerph-19-11418-f001:**
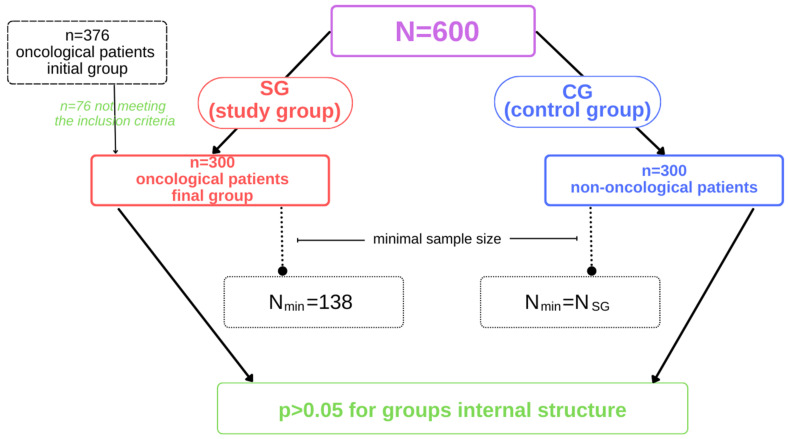
Participants in the study.

**Table 1 ijerph-19-11418-t001:** Questionnaire responses of the study group (SG) and control group (CG) (N = 600).

Anxiety Level by GAD-7	SG	CG
Lack of anxiety	19%	45%
Mild anxiety	12%	30%
Moderate anxiety	10%	10%
Severe anxiety	15%	4%
Generalized anxiety (greater than 10 points)	**44%**	**11%**

**Table 2 ijerph-19-11418-t002:** FCV-19S scale responses in the oncology patients (SG) and control group (CG) (N = 600).

FCV-19S * Scale Discriminator	Group	Strongly Disagree	Rather Disagree	No Opinion	Rather Agree	Strongly Agree
I am very afraid of SARS-CoV-2 (coronavirus)	SG	7%	11%	23%	35%	24%
	CG	20%	23%	43%	11%	3%
	Total	14%	17%	33%	23%	14%
I feel anxious when I think about coronavirus	SG	9%	9%	19%	31%	32%
	CG	26%	17%	29%	17%	11%
	Total	18%	13%	24%	24%	22%
My hands sweat when I think about coronavirus	SG	23%	19%	29%	15%	14%
	CG	48%	31%	9%	13%	9%
	Total	36%	25%	19%	14%	12%
I’m afraid of losing my life due to coronavirus	SG	8%	10%	33%	29%	20%
	CG	34%	27%	17%	9%	13%
	Total	21%	19%	25%	19%	17%
When I watch the news and learn about coronavirus-related stories on social media, I get nervous or anxious	SG	9%	11%	23%	33%	24%
	CG	22%	23%	43%	9%	3%
	Total	16%	17%	33%	21%	14%
I can’t sleep because I’m worried about getting infected with coronavirus	SG	31%	27%	9%	14%	19%
	CG	42%	31%	19%	7%	1%
	Total	37%	29%	14%	11%	10%
My heart beats rapidly when I think of coronavirus infection	SG	23%	19%	19%	17%	22%
	CG	34%	31%	21%	13%	1%
	Total	29%	25%	20%	15%	12%

* Each scale discriminator is scored from 1 to 5, where 5 indicates complete agreement with the statement given.

**Table 3 ijerph-19-11418-t003:** FCV-19S scale scores in oncology patients (SG) and control group (CG) (N = 600).

FCV-19S * Scale Discriminator	Group	X	SD	MIN	MAX	Me	Mo	t	*p*-Value
I am very afraid of SARS-CoV-2 (coronavirus)	SG	4.1	0.9	1	5	3	4	11.298	0.001
	CG	2.8	0.7	1	5	3	3		
	Total	3.4	0.8	1	5	3	3		
I feel anxious when I think about coronavirus	SG	3.5	0.5	1	5	3	4	10.986	0.001
	CG	2.2	0.7	1	5	3	2		
	Total	2.8	0.7	1	5	3	3		
My hands sweat when I think about coronavirus	SG	1.9	0.9	1	5	3	2	12.862	0.001
	CG	1.2	0.9	1	5	3	2		
	Total	1.5	0.9	1	5	3	2		
I’m afraid of losing my life due to coronavirus	SG	2.5	0.7	1	5	3	2	10.632	0.001
	CG	1.2	0.5	1	5	3	2		
	Total	1.7	0.7	1	5	3	2		
When I watch the news and learn about coronavirus-related stories on social media, I get nervous or anxious	SG	3.3	0.5	1	5	3	3	10.256	0.001
	CG	1.5	0.5	1	5	3	2		
	Total	2.5	0.5	1	5	3	2		
I can’t sleep because I’m worried about getting infected with coronavirus	SG	2.5	0.3	1	5	3	3	11.753	0.001
	CG	1.1	0.5	1	5	3	1		
	Total	1.4	0.4	1	5	3	1		
My heart beats rapidly when I think of coronavirus infection	SG	2.3	0.9	1	5	3	2	12.765	0.001
	CG	1.1	0.5	1	5	3	1		
	Total	1.6	0.7	1	5	3	1		
Raw score (points)	SG	20.1 ± 4.7						11.824	0.001
	CG	10.6 ± 4.3							
	Total	14.9 ± 4.5							
Recalculated result (%)	SG	57.4							
	CG	30.3							
	Total	42.6							
Interpretation	SG	Moderate level of anxiety							
	CG	Low anxiety							
	Total	Low anxiety							

Legend: X—mean; SD—standard deviation; MIN—minimum value; MAX—maximum value; Me—median; Mo—modal; t—statistical test; *p*—probability. * Each scale discriminator is scored from 1 to 5, where 5 indicates complete agreement with the statement given.

**Table 4 ijerph-19-11418-t004:** The most frequently reported reasons for anxiety in the study group (N = 600).

Reason for Anxiety	SG	CG	Total	t	*p*-Value
Death alone	82%	54%	68%	12.866	0.023
Complications	46%	56%	51%	13.001	0.013
Treatment delay	62%	38%	50%	10.581	0.001
Income loss	38%	32%	35%	10.549	0.002
Separation	36%	26%	31%	12.987	0.011

Legend: t—statistical test; *p*—probability.

## Data Availability

The data presented in this study are available on request from the corresponding author.

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
