# Peer review of "The Level of COVID-19 Anxiety among Oncology Patients in Poland"

_ijerph, 2022, doi:10.3390/ijerph191811418_

Round 1

Reviewer 1 Report (Previous Reviewer 1)

Yes, this revised paper has been improved and precisely revised!

Author Response

Dear Reviewer,

Thank you for your favorable review of the paper. The changes made in the text are marked in red. A new division of sections in the methodology has been applied, the paper has been improved editorially and linguistically, the conclusions have been shortened, and a graphic describing the sampling has been added.
With best regards, Authors

Reviewer 2 Report (New Reviewer)

Manuscript: The Level of COVID-19 Anxiety Among Oncology Patients in Poland is interesting, but some aspects should be improved.

Thank you very much for your work.

Methodology

It should be include a flowchart with the selection of patients, 

The sample calculation should not be into the patients characteristics.

Change the title research tools by outcome measurements or similary

The conclusions are very extensive

Author Response

Dear Reviewer,

Thank you for your favorable review of the paper. The changes made in the text are marked in red. A new division of sections in the methodology has been applied, the paper has been improved editorially and linguistically, the conclusions have been shortened, and a graphic describing the sampling has been added.
With best regards, Authors

This manuscript is a resubmission of an earlier submission. The following is a list of the peer review reports and author responses from that submission.

Round 1

Reviewer 1 Report

This paper is a unique study of COVID-19 anxiety symptoms in oncology patients in Poland. Significant results have been obtained, including comparisons with controls. Patients with oncology have been shown to be more anxious about COVID-19 than those without. However, at the same time, some percentage of the tumor patients do not feel anxious. What could be the reason for this? Please add that consideration.

Reviewer 2 Report

In this manuscript, the authors investigated the level of covid-19 anxiety among oncology patients in Poland. The topic is timely, however, as an epidemiological study, the manuscript suffers from several fatal issues that limit the scientific soundness of the study.

1), the sample that the authors used is unlikely to be representative of Poland.

2), it is unclear if the sample size is sufficient for a point estimation of the prevalence of covid-19 anxiety.

3), the study lacks multiple variables analysis to addressing many potential confounding factors.

Other issues:

abstract needs a background section; introduction first paragraph needs citations.